# Physics Aware Neural Networks for Unsupervised Binding Energy Prediction

**Ke Liu** [1]   **Hao Chen** [1]   **Chunhua Shen** [1]

## Abstract

Developing models for protein-ligand interactions holds substantial significance for drug discovery. Supervised methods often failed due to the lack of labeled data for predicting the protein-ligand binding energy, like antibodies. Therefore, unsupervised approaches are urged to make full use of the unlabeled data. To tackle the problem, we propose an efficient, unsupervised protein-ligand binding energy prediction model via the conservation of energy (CEBind), which follows the physical laws. Specifically, given a protein-ligand complex, we randomly sample forces for each atom in the ligand. Then these forces are applied rigidly to the ligand to perturb its position, following the law of rigid body dynamics. Finally, CEBind predicts the energy of both the unperturbed complex and the perturbed complex. The energy gap between two complexes equals the work of the outer forces, following the law of conservation of energy. Extensive experiments are conducted on the unsupervised protein-ligand binding energy prediction benchmarks, comparing them with previous works. Empirical results and theoretic analysis demonstrate that CEBind is more efficient and outperforms previous unsupervised models on benchmarks.

## 1. Introduction

Predicting protein-ligand binding is crucial to the discovery of drugs (Zheng et al., 2020; Ballester & Mitchell, 2010; de Freitas & Schapira, 2017). A protein is highly likely to be bonded with a ligand if its conformation, *i.e.*, quaternary structure, holds a high affinity or low binding energy (Chothia & Janin, 1975). Therefore, the protein-ligand binding prediction comes down to the affinity or energy prediction of the protein complex. The ligands encompass proteins, such as antibodies, and small molecules. In the case of small molecules, plenty of labeled samples exist; thus, supervised methods are quite appropriate (Öztürk et al., 2018; Monteiro et al., 2022; Pei et al., 2023; Lu et al., 2022). However, for ligands like antibodies, few samples with an affinity label are available, which causes the failure of supervised approaches.

To address the problem of data scarcity, Feng et al. (2024) constructed a self-supervised reconstruction task in their BindNet to learn binding representations. BindNet does not provide the binding affinity but requires subsequent fine-tuning on downstream tasks. Furthermore, in DSMbind, Jin et al. (2024) proposed the unsupervised protein-ligand binding energy prediction problem. The problem can be defined as given only the quaternary structure of protein-ligand complexes without any affinity labels for training, the model predicts the binding energy. Model performance is evaluated in terms of the correlation between predicted binding energy and ground truth. DSMbind solves the problem by considering the log-likelihood of SE(3) denoising score matching as binding affinity. They train an energy model by matching the forces, *i.e.*, the gradient of the predicted energy with respect to atom coordinates, with the likelihood. Specifically, they perturbed the protein of the complex with a randomly sampled rotation and translation. Then they calculate the rotation caused by the predicted forces and match the predicted force and rotation with the sampled translation and rotation, respectively. However, DSMBind made some approximations, and training a DSMBind model is memory-intensive.

In this work, we propose a memory-efficient and physics-aware training framework dubbed **C**onservation of **E**nergy based **Bind**ing (**CEBind**), which tackles the problem of unsupervised protein-ligand binding energy prediction via the law of conservation of energy. We focus on the unsupervised protein-ligand binding prediction, including antibodies and small molecules, following the settings of DSMBind (Jin et al., 2024). Specifically, given a protein-ligand complex, we randomly sample forces for each atom in the ligand, which is then applied rigidly to the ligand following the rigid dynamics. Then the energy model predicts the energy of both the perturbed and unperturbed complexes. According to the energy conservation law, the difference between the two energies is equal to the work of the outer forces. Besides, we also match the gradient of the energy with respect

[1]Zhejiang University, Hangzhou, China. Correspondence to: Hao Chen <haochen.cad@zju.edu.cn>.

*Proceedings of the 42nd International Conference on Machine Learning*, Vancouver, Canada. PMLR 267, 2025. Copyright 2025 by the author(s).

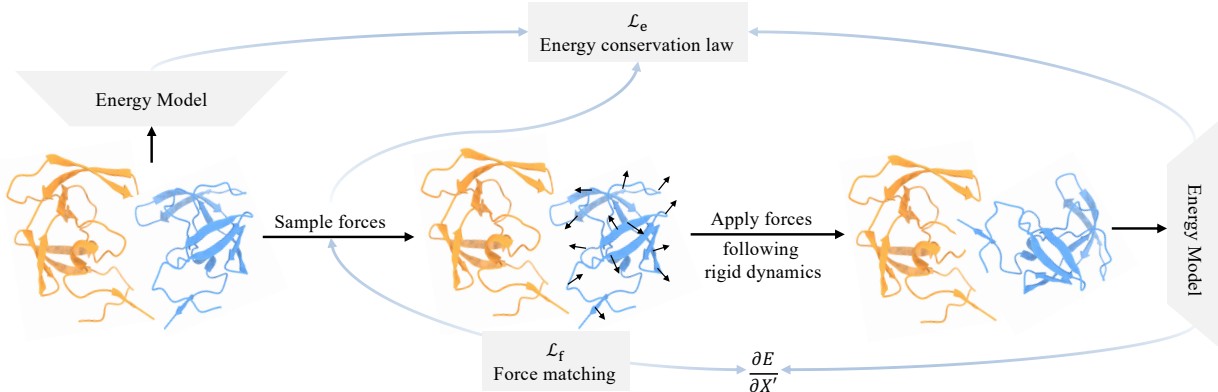

*Figure 1.* Overview of CEBind framework: given a protein-ligand complex, CEBind first samples forces for each atom in the ligand. Then the forces are applied to the ligand following rigid dynamics, to get a perturbed complex. Next, the energy of both the unperturbed and perturbed complexes is predicted by the energy model. Finally, we match the gradient with respect to atom coordinates with inner forces and follow the law of conservation of energy to match the work of outer forces and energy gap.

to atom coordinates with the inner forces. CEBind has the following features:

1. Physics-aware. CEBind follows the law of conservation of energy and rigid dynamics.

2. Memory-efficient. The process of calculating the rigid dynamic of the complex does not require a gradient.

3. Better performance. Extensive experiments are conducted, and empirical results demonstrate that CEBind outperforms previous models.

## 2. Related Works

### 2.1. Protein Molecule Binding Prediction

A lot of approaches have been proposed for protein-ligand binding prediction (Lu et al., 2022; Jin et al., 2024; Somnath et al., 2021; Zhang et al., 2023; Wang et al., 2022; Dittrich et al., 2018). For supervised methods, graph neural networks are often constructed to encode protein-ligand complexes into a representation vector, and a subsequent neural network is performed to predict the binding affinity (Lu et al., 2022; Somnath et al., 2021; Zhang et al., 2023; Wang et al., 2022). Unsupervised approaches based on physics often require expensive computation (Miller III et al., 2012). The most related works to our CEBind are DrugScore (Dittrich et al., 2018) and DSMBind (Jin et al., 2024). DrugScore predicts the energy of each atom pair independently, using atom types and distances. DSMBind further includes information on the protein-ligand complex context. Our CEBind learns a more physics-aware and efficient energy model.

### 2.2. Antibody-antigen Binding Prediction

Data scarcity is extremely high in antibody-antigen binding prediction. Only 566 samples exist in the largest binding affinity dataset of antibodies, SAbDab (Schneider et al., 2022). Few works focus on supervised antibody-antigen binding prediction. Only Myung et al. (2022) predicted the antibody-antigen binding affinity with expensive handcraft feature engineering. Jin et al. (2024) compared their DSM-Bind with Atom3d model (Townshend et al., 2021) and Frame Average model (Puny et al., 2022) they implemented for supervised antibody-antigen binding affinity prediction. Antibody design models often include modules for modeling the antibody, antigen, and the interaction between them (Viswanath et al., 2013; Alford et al., 2017). We also adapt the antibody design model architecture and compare it with these unsupervised approaches.

### 2.3. Unsupervised Binding Energy Prediction

Jin et al. (2024) introduced the task of unsupervised protein-ligand binding energy prediction for both protein-small molecules and antibody-antigens. Operating only with the quaternary structures, namely, the prevailing protein-ligand complexes, a model is trained to predict the complexes' binding affinity. Due to the non-availability of binding affinity labels for training, the model's performance is evaluated through the correlation between the predicted affinity and ground truth affinity. Our work parallels the task setting in DSMBind (Jin et al., 2024). However, we put forth a physics-aware and more streamlined training framework.

## 3. Preliminaries and Notations

**Protein-Ligand structures.** A protein-ligand complex is described as its atom features $\boldsymbol{A} = [\boldsymbol{A}_{\mathrm{P}}, \boldsymbol{A}_{\mathrm{L}}]$ and posi-

tions $X = [X_P, X_L]$, where $A_P$ and $X_P$ denote the protein and $A_L$ and $X_L$ denote the ligand, *i.e.*, $C = [A, X]$. The atom features consist of individual atom types $A = [a_1, a_2, \cdots, a_n]$, where $a_i \in \mathcal{C}^{95}$ denotes the atom type and $n$ denotes the number of atoms. $X = [x_1, x_2, \cdots, x_n]$, where $x_n \in \mathbb{R}^3$ denotes the 3D atom position. The proteins in protein-ligand complexes can also be described with their amino acids, *i.e.*, $P = [p_1, p_2, \cdots, p_3]$, where each amino acid consists of atoms $p_i = [a_k, a_{k+1}, \cdots, a_{k+m}]$. The small molecules in the complexes can be described with their molecule graphs, *i.e.*, $G = (A_P, E, X_P)$, where $E$ denotes the edges between atoms.

**Problem formulation.** Given only the structure of a protein-ligand complex $C = (A, X)$, an unsupervised protein-ligand binding energy model is trained to predict the pseudo-binding energy of the complex $E(A, X)$. Since the object of training does not involve the ground truth of binding energy, the metrics for evaluation and testing only include the correlation between predicted pseudo-binding energy and the ground truth.

**Rotation.** Following Jin et al. (2024), we use the rotation vector $\omega = [\omega_x, \omega_y, \omega_z]$ and the rotation matrix $R_\omega$ for rotating the atoms. The rotation matrix $R_\omega$ is applied to the atom coordinates $x_i$ through:

$$R_\omega x_i = x_i + c_1 \omega \times x_i + c_2 \omega \times (\omega \times x_i), \quad \text{(Rotation)}$$

where

$$c_1 = \frac{\sin \|\omega\|}{\|\omega\|}, \quad c_2 = \frac{1 - \cos \|\omega\|}{\|\omega\|^2}. \quad (1)$$

## 4. Methodology

In this section, we describe our binding energy prediction method, which follows the law of conservation of energy and rigid dynamics. The overview of our CEBind is shown in Fig. 1. Given a protein-ligand complex, we sample forces for each atom in the ligand, which are then applied to the ligand following rigid dynamics. Finally, we calculate the energy of the perturbed complex and the unperturbed complex. The energy gap between them equals the work of the outer forces, and the gradient of the perturbed complex with respect to atom coordinates matches the inner forces.

### 4.1. Rigid Dynamics

Given a protein-ligand complex, $C = (A, X)$, we randomly sample forces $f \in \mathbb{R}^3$ as the combined forces for each atom in the ligand. Then we apply these forces to the ligand, following rigid dynamics to preserve the structure of ligands. To remove the effect of kinetic energy, we perform the force $f$ during $(0, \frac{1}{2}\Delta t)$, and perform the force $-f$ during $(\frac{1}{2}\Delta t,$

$\Delta t)$, where $\Delta t$ is the total duration that the forces apply to the ligand.

The translation of each atom in the ligand consists of three parts: (1) the translation of the ligand caused by combined forces; (2) the ligand rotation, which causes the translation of all the atoms in the ligand; and (3) the rotation of bonds that can rotate.

The translation of ligands caused by combined forces can be easily calculated according to Newton's laws of motion, as follows:

$$\Delta X_{L1} = \frac{1}{2} \frac{\sum_{i=0}^{N} f_i}{M} \Delta t^2, \quad (2)$$

where $N$ and $M$ indicate the number of atoms in the ligand and the mass of the ligand, respectively.

The translation caused by the ligand rotation is obtained through rigid dynamics. First of all, we calculate the inertia matrix $I_N \in \mathbb{R}^{3 \times 3}$ and torque $\tau \in \mathbb{R}^3$ of the ligand as follows:

$$I_N = \sum_{i=1}^{N} \|x_i - \mu\|^2 - (x_i - \mu)(x_i - \mu)^\top, \quad (3)$$

$$\tau = \sum_{i=1}^{N} (x_i - \mu) \times f_i, \quad (4)$$

where $\mu$ is the center of the ligand and calculated as follows:

$$\mu = \frac{1}{N} \sum_{i}^{x_i \in X_L} x_i. \quad (5)$$

Then we apply the torque to the ligand and get the atom translation caused by the ligand rotation as follows:

$$\Delta X_{L2} = \text{Rot}(\frac{1}{2} I_N^{-1} \tau \Delta t^2, X_L - \mu) - X_L, \quad (6)$$

where the operation Rot is described in Eq. Rotation. More details can be found in the Appendix. A.1.

For the bonds that can rotate, we also model the rotation of them by taking the fixed atoms as the origin and performing rotating on it. Here we take the side chain for example. The side chain consists of its atom types $A_S$ and atom positions $X_S$, *i.e.*, $C_S = (A_S, X_S)$ without the backbone atoms, *i.e.*, $C_S \subset C$. The fixed position for the side chain rotation is $\alpha$-carbon, the positions of which are denoted as $X_{S\alpha}$. Similarly, we have the inertia matrix and torque of the side chains as follows:

$$I_S = \sum_{i=1}^{J} \|x_i - X_{S\alpha}\|^2 - (x_i - X_{S\alpha})(x_i - X_{S\alpha})^\top, \quad (7)$$

$$\tau_S = \sum_{i=1}^{J} (x_i - X_{S\alpha}) \times f_i, \quad (8)$$

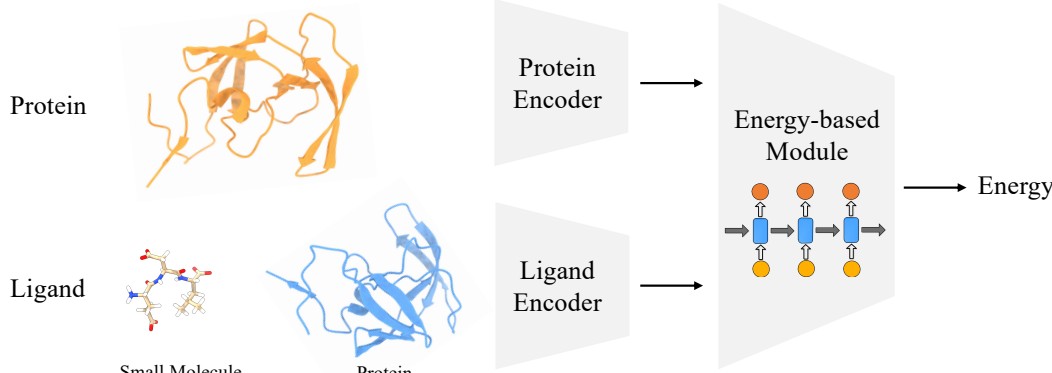

*Figure 2.* Model architecture: our energy model consists of a protein encoder, a ligand encoder, and an energy-based module. The protein encoder and the ligand encoder learn representations for each pair of protein-ligand complexes. Energy-based module model the interaction between ligand and protein and predict their binding energy.

The translation caused by the side chain rotation is obtained as:

$$\Delta \boldsymbol{X}_{\mathrm{S}} = \mathrm{Rot}(\frac{1}{2}\boldsymbol{I}_{\mathrm{S}}^{-1}\boldsymbol{\tau}_{\mathrm{S}}\Delta t^2, \boldsymbol{X}_{\mathrm{S}} - \boldsymbol{X}_{\mathrm{S}\alpha}) - \boldsymbol{X}_{\mathrm{S}}. \quad (9)$$

Finally, the translation of each atom is formulated through combining all the translations in Eq. 2. Eq. 6, and Eq. 9 as follows:

$$\Delta \boldsymbol{X}_{\mathrm{L}} = \Delta \boldsymbol{X}_{\mathrm{L}1} + \Delta \boldsymbol{X}_{\mathrm{L}2} + [0, \Delta \boldsymbol{X}_{\mathrm{S}}], \quad (10)$$

where $[0, \Delta \boldsymbol{X}_{\mathrm{S}}]$ indicates the noise on the backbone atoms is 0 and the noise on the side chain atoms is $\Delta \boldsymbol{X}_{\mathrm{S}}$. The perturbed protein-ligand complex is $\tilde{C}(\boldsymbol{A}, \tilde{\boldsymbol{X}}) = \tilde{C}(\boldsymbol{A}, [\boldsymbol{X}_{\mathrm{P}}, \tilde{\boldsymbol{X}}_{\mathrm{L}}])$, where $\tilde{\boldsymbol{X}}_{\mathrm{L}}$ is perturbed with Eq. 10 as follows:

$$\tilde{\boldsymbol{X}}_{\mathrm{L}} = \boldsymbol{X}_{\mathrm{L}} + \Delta \boldsymbol{X}_{\mathrm{L}} = \boldsymbol{X}_{\mathrm{L}} + \Delta \boldsymbol{X}_{\mathrm{L}1} + \Delta \boldsymbol{X}_{\mathrm{L}2} + [0, \Delta \boldsymbol{X}_{\mathrm{S}}]. \quad (11)$$

The details of the force-performing process are illustrated in appendix. A.2.

### 4.2. Energy Conservation Law

According to the law of conservation of energy, the total energy of an isolated system remains constant and the energy change in the system is equal to the work done on that system from the outside. Taking the protein-ligand complex as an isolated system, the binding energy of the complex remains constant. After perturbing the ligand with the sampled forces $f$, the changes in the binding energy of the complex equal the work of the outer forces $\boldsymbol{f}_{outer}$. Here, we model the inner force between the ligand and the protein to be proportional to their distance, *i.e.* $\boldsymbol{f}_{inner} \propto (\boldsymbol{X}_t - \boldsymbol{X}_0)$ and $\boldsymbol{f} = \boldsymbol{f}_{inner} + \boldsymbol{f}_{outer}$ (Without loss of generalization, any other force field modeling can be applied here without changing our framework). With a scaler $k$, we have $\boldsymbol{f}_{inner} = k(\boldsymbol{X}_t - \boldsymbol{X}_0)$.

According to Jammer (1999), work is the inner product of force and displacement. The work of outer forces is obtained through the inner product of force and displacement, as follows:

$$\boldsymbol{W} = \int_{\boldsymbol{X}_0}^{\boldsymbol{X}_0 + \Delta \boldsymbol{X}_{\mathrm{L}}} \boldsymbol{f}_{outer}\, d\boldsymbol{X} = \frac{1}{2}k\Delta \boldsymbol{X}_{\mathrm{L}}{}^2, \quad (12)$$

where the translation $\boldsymbol{X}_{\mathrm{L}}$ is obtained through Eq. 10. The derivation of Eq. 12 can be found in the Appendix. A.3.

The energy of the protein-ligand complex $C(\boldsymbol{A}, \boldsymbol{X})$ is estimated by the energy model $kE_\theta(\boldsymbol{A}, \boldsymbol{X})$. For the perturbed complex $C(\boldsymbol{A}, \tilde{\boldsymbol{X}})$, the energy is $kE_\theta(\boldsymbol{A}, \tilde{\boldsymbol{X}})$. Then we match the change in energy with the work done by the outer forces as follows:

$$\mathcal{L}_{\mathrm{e}} = \mathbb{E}[\|kE_\theta(\boldsymbol{A}, \tilde{\boldsymbol{X}}) - kE_\theta(\boldsymbol{A}, \boldsymbol{X}) - \boldsymbol{W}\|^2] \quad (13)$$

$$= k\mathbb{E}[\|E_\theta(\boldsymbol{A}, \tilde{\boldsymbol{X}}) - E_\theta(\boldsymbol{A}, \boldsymbol{X}) - \frac{1}{2}\Delta \boldsymbol{X}_{\mathrm{L}}{}^2\|^2]. \quad (14)$$

### 4.3. Force Score Matching

The gradients of an energy with respect to atom coordinates are forces that move the ligand to a position with a lower energy. Therefore, the predicted inner forces by the energy model are calculated as follows:

$$\tilde{\boldsymbol{f}}_{inner} = \frac{\partial kE_\theta(\boldsymbol{A}, \tilde{\boldsymbol{X}})}{\partial \tilde{\boldsymbol{X}}_{\mathrm{L}}} \quad (15)$$

Following Jin et al. (2024), we also calculate the gradients of the perturbed complex as the force to match the inner forces via a denoising score matching as follows:

$$\mathcal{L}_{\boldsymbol{f}} = \mathbb{E}[\|\tilde{\boldsymbol{f}}_{inner} - \nabla_{\boldsymbol{f}_{inner}} \log p(\boldsymbol{f}_{inner})\|^2], \quad (16)$$

$$= k\mathbb{E}[\|\frac{\partial E_\theta(\boldsymbol{A}, \tilde{\boldsymbol{X}})}{\partial \tilde{\boldsymbol{X}}_{\mathrm{L}}} + \frac{\Delta \boldsymbol{X}_{\mathrm{L}}}{\sigma^2}\|^2] \quad (17)$$

---

**Algorithm 1** Training procedure (single data point)

---

**Require:** A protein-ligand complex $C(\boldsymbol{A}, \boldsymbol{X})$

1: Sample a noise level $\sigma$.
2: Sample forces for each atom in ligand $\boldsymbol{f} \sim \mathcal{N}(0, \sigma^2 \boldsymbol{I})$.
3: Calculate the translation of each atom $\Delta \boldsymbol{X}_\mathrm{L}$ with the forces via Eq. 10.
4: Calculate the work of outer forces $\boldsymbol{W}$ via Eq. 12.
5: Apply the noise to the original complex to get the perturbed complex $\hat{C} = (\boldsymbol{A}, \tilde{\boldsymbol{X}})$ via Eq. 11.
6: Predict the energy of both complexes $E_\theta(\boldsymbol{A}, \boldsymbol{X})$ and $E_\theta(\boldsymbol{A}, \tilde{\boldsymbol{X}})$ via energy model.
7: Minimize the energy loss $\mathcal{L}_\mathrm{e}$ and force matching loss $\mathcal{L}_\mathrm{f}$ from Eq. 13 and Eq. 16.

---

Different from DSMBind (Jin et al., 2024), the self-supervision is on each atom instead of the combined translation and rotation of the whole ligand (Song et al., 2024a; Yim et al., 2023; Song et al., 2024b). More details can be found in Appendix. A.4.

### 4.4. Model Architecture

The whole model consists of a protein encoder, a ligand encoder, and an energy-based module, as shown in Fig. 2. We use the encoders to encode the protein and ligand into representation vectors. Then the vectors are input into the energy model to model the interactions between protein and ligand. Finally, the energy-based module outputs the binding energy of the complex. The training procedure for our CEBind is described in the Algorithm 1. The first five steps can be done in the dataloader and these operations do not require gradient calculation in the backward propagation. Therefore, CEBind is more memory-efficient. Besides, CEBind follows physics laws, including rigid dynamics and the law of energy conservation.

## 5. Experiment & Discussion

We trained the energy model with our CEBind framework. We first compare the performance of our CEBind with other approaches. Then we analyze our CEBind in terms of efficacy. Finally, we propose an atom-only variant of CEBind to explore the generalization of the unsupervised binding energy prediction approaches.

### 5.1. Setup

**Dataset.** The datasets used in this work consist of **protein-small molecule** dataset and **antibody-antigen** dataset following (Jin et al., 2024). The **protein-small molecule** dataset contains 4806 protein-ligand complexes from PDBbind V2020 database for training (Su et al., 2018), 357 complexes randomly sampled from PDBbind in (Stärk et al.,

2022) for evaluation, and 258 complexes from the PDBbind core set with labels of binding energy (Su et al., 2018) for test. The protein is cropped to its binding pocket in the protein-small molecule dataset. The pocket is defined as the residues with a distance less than 10Å from the ligand. The **antibody-antigen** dataset includes 3416 antibody-antigen complexes from the structural antibody database (SAbDab) (Schneider et al., 2022) for training, 116 complexes from CSM-sb (Myung et al., 2022) for evaluation, and 566 complexes with labels of binding affinity from SAbDab for test. All test samples never exist in the training datasets. Each entry in the training sets contains only the protein-ligand complexes. The binding affinity labels exist only in the validation and test sets.

**Evaluation metrics.** We mainly employ the **Pearson correlation coefficient** between the ground truth protein-ligand binding affinity and the predicted binding energy, which is calculated as follows:

$$r_p = \frac{\sum_i (E_\theta(\boldsymbol{A}, \boldsymbol{X})_i - \overline{E_\theta(\boldsymbol{A}, \boldsymbol{X})})(y_i - \overline{y})}{\sqrt{\sum_i (E_\theta(\boldsymbol{A}, \boldsymbol{X})_i - \overline{E_\theta(\boldsymbol{A}, \boldsymbol{X})})^2 \sum_i (y_i - \overline{y})^2}}, \tag{18}$$

where $y_i$ is the ground truth binding energy of the $i$-th protein-ligand complex. We train the model with five different random seeds and report the mean and standard deviation of $r_p$. The complete results can be found in the Appendix. C.1.

**Compared approaches.** For protein-small molecule binding, we mainly compare our CEBind with physics-based potentials (including Glide (Friesner et al., 2006), AutoDock$_\mathrm{vina}$ (Eberhardt et al., 2021), MM/GBSA (Miller III et al., 2012), and DrugScore$_{2018}$ (Dittrich et al., 2018)), unsupervised models (including standard Gaussian denoising score matching (Gauss DSM) (Jin et al., 2024), Contrastive learning (Chen et al., 2020), and DSMBind (Jin et al., 2024)), and supervised approaches (including IGN (Jiang et al., 2021) and PLANET (Zhang et al., 2023)) following Jin et al. (2024). The supervised approaches were trained with over 19,000 samples with binding affinity from the PDBBind database. For antibody-antigen binding, we also compare our CEBind with physics-based models (including all the approaches in the CCharPPI server (Moal et al., 2015)), unsupervised models (including contrastive learning (Chen et al., 2020), Gauss DSM (Jin et al., 2024), ESM-1v (Meier et al., 2021), ESM-IF (Hsu et al., 2022), and DSMBind (Jin et al., 2024)), and supervised approaches, frame-averaging neural network (FANN) (Puny et al., 2022). FANN$_\mathrm{ab}$ is trained on extra 273 antibody-antigen samples with binding affinity labels from protein-protein binding mutation database (SKEMPI) (Jankauskaitė et al., 2019). FANN$_\mathrm{transfer}$ is trained on extra 5427 samples from SKEMPI (Jankauskaitė et al., 2019) and then fine-

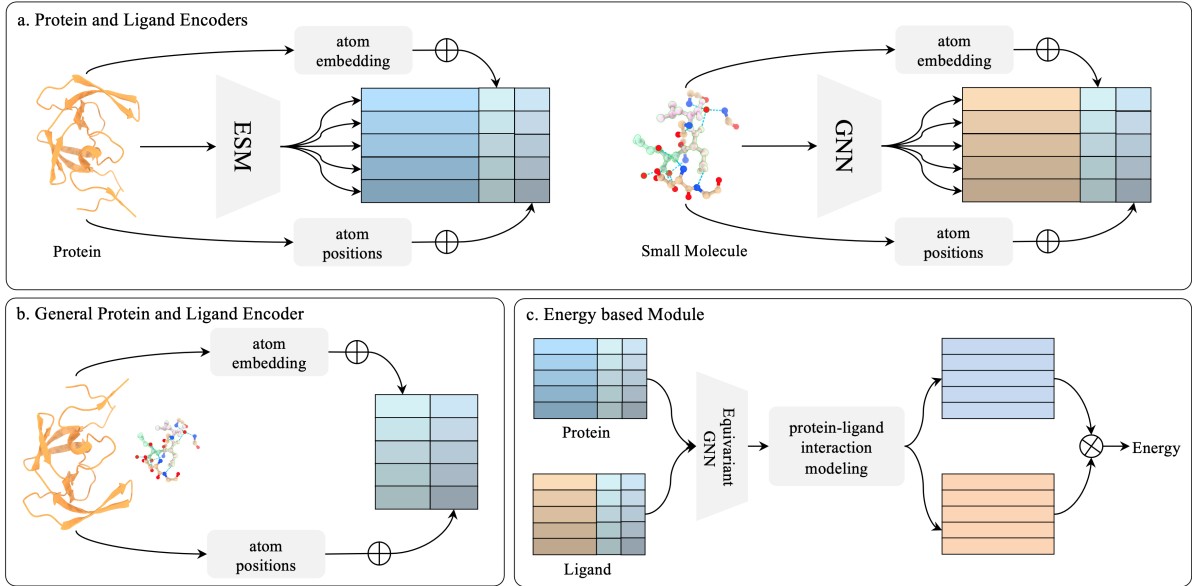

*Figure 3.* Implementation of CEBind. (a) Protein encoder and small molecule encoder. The protein encoder consists of the atom embedding module, ESM module, and atom positions. The small molecule encoder contains a graph neural network (GNN), atom embedding module, and atom positions. They both output the combination of the three features. ESM module outputs the representation of each residue in the protein. (b) A general encoder for both small molecules and proteins. The general encoder outputs the representation of ligands with a combination of their atom embedding and atom positions. (c) The architecture of the energy-based model. The energy model includes an equivariant GNN and a protein-ligand interaction modeling module. $\otimes$ indicates the multiplication.

tuned on the 273 samples. The comparison mainly follows DSMBind (Jin et al., 2024).

### 5.2. Implementation Details

The implementation of our CEBind is illustrated in Fig. 3. We implement two variants of the ligand encoder. Following previous works (Jin et al., 2024; Schneider et al., 2022; Eberhardt et al., 2021), we first implement the ligand encoder for protein and small molecules, respectively, as shown in Fig. 3(a). The protein encoder consists of an atom embedding module to represent each atom in the protein as a vector and an ESM module to model the interaction inside the protein and embed each residue into a vector. Then, the two vectors and atom positions are concatenated to represent each atom as follows:

$$\boldsymbol{R}_{\mathrm{LP}} = \mathrm{Embed}(\boldsymbol{A}_{\mathrm{L}}) \oplus \mathrm{ESM}(\boldsymbol{P}_{\mathrm{L}}) \oplus \boldsymbol{X}_{\mathrm{L}}, \quad (19)$$

where $\oplus$, $\boldsymbol{A}_{\mathrm{L}}$, and $\boldsymbol{P}_{\mathrm{L}}$ indicate the concatenation, atom representation, and protein residues.

The small molecule encoder consists of a GNN to model the interaction inside the molecule and embed each atom into a vector. With the atom representation from the atom embedding module and the atom positions, the atom representation of the small molecule is obtained through:

$$\boldsymbol{R}_{\mathrm{LM}} = \mathrm{Embed}(\boldsymbol{A}_{\mathrm{L}}) \oplus \mathrm{GNN}(G) \oplus \boldsymbol{X}_{\mathrm{L}}. \quad (20)$$

To explore the gap between small molecules and proteins, we implement a general ligand encoder for both small molecules and proteins as shown in Fig. 3(b). The general encoder consists of an atom embedding module. By concatenating the atom positions and atom embedding as

$$\boldsymbol{R}_{\mathrm{L}} = \mathrm{Embed}(\boldsymbol{A}_{\mathrm{L}}) \oplus \boldsymbol{X}_{\mathrm{L}}, \quad (21)$$

the general encoder can be used for both proteins and small molecules.

The energy-based module contains an equivariant graph neural network and a protein-ligand interaction modeling module following Jin et al. (2024) as shown in Fig. 3(c). With the two modules, the representations for each atom in the ligand is obtained as $[\boldsymbol{R}_{\mathrm{L}}', \boldsymbol{R}_{\mathrm{P}}'] = \phi([\boldsymbol{R}_{\mathrm{L}}, \boldsymbol{R}_{\mathrm{P}}])$, where $\boldsymbol{R}_{\mathrm{P}}$ is obtained through Eq. 19 or Eq. 21. Then a multiplication is performed between the atom representation $\boldsymbol{R}_{\mathrm{L}}'$ and $\boldsymbol{R}_{\mathrm{P}}'$ to get the energy of each atom pair. The binding energy of a complex is the sum of the energies of all the atom pairs within a distance threshold.

### 5.3. Experimental Results

The comparison of CEBind with previous methods demonstrates the efficacy of CEBind. Then we demonstrate the efficiency of CEBind both theoretically and empirically. Finally, we conduct experiments with the general atom-only protein and ligand encoder to explore the gap between small

*Table 1.* Pearson correlation on protein-small molecule and antibody-antigen datasets. * denotes that the results are reproduced from their official code. $r_p$ denotes the Pearson correlation. The **best** results in each category of approaches are labeled in bold.

| | Protein-small molecule | | Antibody-antigen | |
|---|---|---|---|---|
| | Method | $r_p \uparrow$ | Method | $r_p \uparrow$ |
| Supervised | PLANET | 0.811 | $FANN_{ab}$ | $0.325 \pm 0.014$ |
| | IGN | **0.837** | $FANN_{transfer}$ | **0.350** $\pm 0.033$ |
| Physics-based | Glide | 0.467 | ZRANK | 0.318 |
| | $Autodock_{vina}$ | 0.604 | CP_PIE | 0.234 |
| | $DrugScore_{2018}$ | 0.602 | $PYDOCK_{vina}$ | 0.248 |
| | MM/GBSA | **0.647** | AP_PISA | **0.323** |
| Unsupervised | | | ESM-1v | 0.024 |
| | | | ESM-IF | 0.024 |
| | Contrastive | $0.625 \pm 0.002$ | Contrastive | $0.308 \pm 0.037$ |
| | Gauss DSM * | $0.636 \pm 0.008$ | Gauss DSM * | $0.360 \pm 0.016$ |
| | DSMBind * | $0.644 \pm 0.003$ | DSMBind * | $0.365 \pm 0.011$ |
| | CEBind | **0.652** $\pm 0.005$ | CEBind | **0.374** $\pm 0.009$ |

*Table 2.* Pearson correlation on protein-small molecule and antibody-antigen datasets. $CEBind_{atom}$ denotes the general encoder for ligands. Cross Val denotes that $CEBind_{atom}$ is trained and tested on different datasets.

| Method | DSMBind | CEBind | $CEBind_{atom}$ | Cross Val |
|---|---|---|---|---|
| Protein-small molecule | 0.644±0.003 | 0.652±0.005 | 0.648±0.003 | 0.646±0.001 |
| Antibody-antigen | 0.365±0.011 | 0.374±0.009 | 0.346±0.004 | 0.320±0.002 |

*Table 3.* Ablation study. Pearson correlation on the protein-small molecule and antibody-antigen datasets. $\mathcal{L}_e$ and $\mathcal{L}_f$ denote the energy conservation and force matching, respectively.

| $\mathcal{L}_e$ | $\mathcal{L}_f$ | ESM | Protein-molecule($r_p \uparrow$) | Antibody-antigen($r_p \uparrow$) |
|---|---|---|---|---|
| ✓ | | ✓ | $0.646 \pm 0.002$ | $0.361 \pm 0.016$ |
| | ✓ | ✓ | $0.646 \pm 0.001$ | $0.362 \pm 0.025$ |
| ✓ | ✓ | | $0.650 \pm 0.004$ | $0.346 \pm 0.010$ |
| ✓ | ✓ | ✓ | $0.652 \pm 0.005$ | $0.374 \pm 0.009$ |

molecules and proteins in the unsupervised protein-ligand binding energy prediction task.

We compare CEBind with previous methods on the protein-small molecule dataset and antibody-antigen dataset. CE-Bind outperforms previous unsupervised models in terms of the Pearson correlation coefficient on both datasets, as shown in Table 1. For protein-small molecule binding energy prediction, supervised methods outperform unsupervised approaches, including CEBind, since they are trained with plenty of labeled data. Although the best physics-based approach, MM/GBSA, is comparable with CEBind, it is computationally intensive.

For antibody-antigen binding energy prediction, unsupervised CEBind outperforms all previous approaches, including supervised models, since the labeled data for supervised model training is scarce. The supervised model

$FANN_{transfer}$ is pre-trained on the 5427 samples from SKEMPI (Jankauskaitė et al., 2019) and fine-tuned with 273 antibody-antigen samples with binding affinity labels. $FANN_{ab}$ is trained directly on the 273 samples with binding affinity labels. The training for supervised models is done because there are only 566 samples with binding affinity labels in SAbDb and all of them are in the test set.

### 5.4. Ablation Study

To evaluate the effectiveness of our conservation of energy and force matching, we conducted an ablation study as shown in Table. 3. Both force matching and conservation of energy are effective, with which CEBind achieves an improvement of 2.2% and 2.1%, respectively. ESM plays an important role in antibody-antigen binding energy prediction. However, it is not essential for protein-small molecules

since atom-scale interaction is more important in protein-small molecule interaction. More visual results can be found in Appendix. C.4.

### 5.5. Discussion

#### 5.5.1. EFFICIENCY OF CEBIND

We mainly compare CEBind with the second-best unsupervised approach, DSMBind, in terms of training time and GPU memory consumption. All the comparisons are performed with the same settings. The average training time of each batch for DSMBind and CEBind is 0.0152 seconds and 0.0116 seconds, respectively. CEBind is faster than DSMBind since the process of performing rigid dynamics is done before training in data processing, *i.e.*, the first five steps in Algorithm 1. The average peak GPU memory utilization for CEBind and DSMBind is 4.89 GB and 6.43 GB, respectively. CEBind requires less GPU memory for training than DSMbind, which sheds light on co-training with other models, like designing a complex with high binding affinity. The low memory requirement of CEBind is due to the fact that fewer operations require gradient calculation. More details can be found in the Appendix. C.2.

#### 5.5.2. BIAS OF FORCE APPLICATION

CEBind applies the sampled forces exclusively to ligands, not to the target protein, which could introduce bias. However, any other forces can be applied to the complex without altering our framework. We also explore the application of forces on both protein and ligand in a complex. The results of applying the forces to both protein and ligand and applying forces only to ligand are comparable on both the protein-ligand and antibody-antigen datasets as shown in Table. 4. On the protein-small molecule dataset, by applying forces on both ligand and protein, the performance of CEBind increases slightly since the side chains are included in the rigid dynamics. On the Antibody-antigen dataset, the performance decreases slightly since too many atoms are included. For both datasets, the results are comparable since all relative positions of the proteins and ligands can be obtained by moving only the ligands if the side chains are not taken into account.

#### 5.5.3. GAP BETWEEN PROTEIN-SMALL MOLECULE AND ANTIBODY-ANTIGEN

To explore the gap between small molecules and proteins as ligands in protein-ligand binding, we implement a general encoder for both protein and small molecule ligands, as shown in Fig. 3(b). We explore the gap through two ways: (1) Evaluate an atom-only model with the general encoder, $CEBind_{atom}$. (2) Train $CEBind_{atom}$ on one dataset and evaluate it on the other dataset. The Pearson correlation results are shown in Table. 2. $CEBind_{atom}$ and CEBind are

*Table 4.* Pearson correlation on protein-small molecule and antibody-antigen datasets. "Only ligand" and "Both" indicate forces applying on the ligand and forces applying on the ligand and protein, respectively.

| Model | Protein-small molecule | Antibody-antigen |
|---|---|---|
| Only ligand | $0.652\pm0.005$ | $0.374\pm0.009$ |
| Both | $0.665\pm0.011$ | $0.362\pm0.015$ |

comparable on the protein-small molecule dataset, while on the antibody-antigen dataset, $CEBind_{atom}$ performs worse than CEBind due to the absence of ESM. Although both GNN and ESM model the interaction inside ligands, ESM is critical for proteins because of its large-scale pre-training, while GNN is not essential for molecules since the interaction modeling is also performed in the energy-based model. Although the performance of $CEBind_{atom}$ decreases on the antibody-antigen dataset, it is still comparable with the best supervised model ($r_p = 0.350$) and better than the best physics-based approach ($r_p = 0.323$).

In the Cross Val, $CEBind_{atom}$ trained on the antibody-antigen dataset is well generalized to the protein-small molecule dataset, with a performance better than all previous unsupervised models. Although generalizing $CEBind_{atom}$ trained on the protein-small molecule dataset to the antibody-antigen dataset is hard since the size of molecules and proteins differs, it is still comparable with the best physics-based model. The size range of samples in the antibody-antigen training set encompasses almost all the sample sizes in the protein-small molecule test set. However, the size ranges of the protein-small molecule training set and the antibody-antigen test set overlap only marginally. More details can be found in Appendix. C.3.

With the results that $CEBind_{atom}$ is generalizable in both datasets, our CEBind and AlphaFold3 (Abramson et al., 2024) come to the same conclusion that a universal approach to all biomolecules, including both proteins and small molecules, is possible.

## 6. Conclusion

In this work, we propose an efficient training framework for unsupervised protein-ligand binding energy prediction via the conservation of energy, named CEBind. By sampling forces on each atom and applying them rigidly to the ligand, CEBind follows physics laws, including rigid dynamics and the energy conservation law. Experiments demonstrate that CEBind is memory-effective and outperforms previous methods. Furthermore, we implement a general encoder for both antibodies and small molecules to explore the gap between them. The generalization of the general encoder

indicates that a potential universal approach exists for all biomolecules, including both proteins and small molecules, which is the same as AlphaFold3.

## Impact Statement

The goal of this work is to advance the field of Machine Learning and Computational Biology. While there are many potential societal consequences of our work, we believe that none of which must be specifically highlighted here.

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

# A. Method Details

*Figure 4.* Force performing process. (1) Calculate the rotation and translation of the whole ligand. (2) For the first rotatable bond, we fix the substructure connected to the bond and calculate the new position with the sample forces according to the kinematic theory. (3) For the second rotatable bond, we repeat the process (2).

## A.1. Rigid Dynamics Derivation

Given a protein-ligand complex, $C = (A, X)$, we randomly sample forces $f \in \mathbb{R}^3$ for each atom in the ligand, Then we apply these forces to the ligand, following the rigid dynamics. The translation of each atom in the ligand consists of three parts: (1) the translation of the ligand caused by combined forces; (2) the ligand rotation, which causes the translation of all the atoms in the ligand; and (3) the rotation of bonds that can rotate.

**(1) Ligand translation:**

$$\Delta X_{\mathrm{L}1} = \frac{1}{2} \frac{\sum_{i=0}^{N} f_i}{M} \Delta t^2. \tag{22}$$

**(2) Ligand rotation:** The translation caused by the ligand rotation is obtained through rigid dynamics. First of all, we calculate the inertia matrix $I_N \in \mathbb{R}^{3 \times 3}$ and torque $\tau \in \mathbb{R}^3$ of the ligand as follows:

$$I_N = \sum_{i=1}^{N} \|x_i - \mu\|^2 - (x_i - \mu)(x_i - \mu)^\top, \tag{23}$$

$$\tau = \sum_{i=1}^{N} (x_i - \mu) \times f_i, \tag{24}$$

where $\mu$ is the center of the ligand and calculated as follows:

$$\mu = \frac{1}{N} \sum_{i}^{x_i \in X_{\mathrm{L}}} x_i. \tag{25}$$

Then we apply the torque to the ligand and the acceleration is:

$$a = I_N^{-1} \tau. \tag{26}$$

with the acceleration, we have the angular displacement as:

$$\omega = \frac{1}{2} I_N^{-1} \tau \Delta t^2. \tag{27}$$

With the rotation equation in Eq. Rotation, we have the atom coordinates as:

$$X_{\mathrm{L}}' = \mathrm{Rot}(\omega, X_{\mathrm{L}} - \mu). \tag{28}$$

Finally, we have the translation of the atoms caused by the rotation as:

$$\Delta X_{\mathrm{L}} = X_{\mathrm{L}}' - X_{\mathrm{L}} = \mathrm{Rot}(\frac{1}{2} I_N^{-1} \tau \Delta t^2, X_{\mathrm{L}} - \mu) - X_{\mathrm{L}}. \tag{29}$$

where the operation Rot is described in Eq. Rotation.

**(3) Side chain rotation.** The derivation of translation caused by the side chain rotation is similar to the ligand rotation as above.

## A.2. Force Performing Process

The process of performing sampled forces is illustrated in Fig. 4. First, we calculate the translation and rotation of the whole ligand. Then for each rotatable bond, we fix the two substructures connected to the bond and calculate the rotation of each substructure as a rigid. Finally, we sum the position changes calculated for all the rotatable bonds and the whole ligand as the final position of a ligand.

## A.3. Work of Outer Forces (the Energy Change).

Given the combined forces $\boldsymbol{f}$, inner force $\boldsymbol{f}_{inner}$, and outer force $\boldsymbol{f}_{outer}$, $\boldsymbol{f} = \boldsymbol{f}_{inner} + \boldsymbol{f}_{outer}$. We perform the $\boldsymbol{f}$ during $(0, \frac{1}{2}\Delta t)$ and $-\boldsymbol{f}$ during $(\frac{1}{2}\Delta t, \Delta t)$. In a small range of $\Delta \boldsymbol{X}$, we model the inner force between the ligand and the protein to be proportional to their distance, *i.e.* $\boldsymbol{f}_{inner} \propto (\boldsymbol{X}_t - \boldsymbol{X}_0)$ (More accurate force field modeling can be adapted here without changing our framework). With a scaler $k$, the force is represented as $\boldsymbol{f}_{inner} = k(\boldsymbol{X}_t - \boldsymbol{X}_0)$. The work of the combined forces is equal to zero since the displacement $\boldsymbol{X}_1$ during $(0, \frac{1}{2}\Delta t)$ equals the displacement $\boldsymbol{X}_2$ during $(\frac{1}{2}\Delta t, \Delta t)$ as follows:

$$\boldsymbol{W}_{combine} = \boldsymbol{f}\boldsymbol{X}_1 + (-\boldsymbol{f}\boldsymbol{X}_2) = 0.$$

Inner and outer forces do work of the same magnitude and in opposite directions as follows:

$$\int_{\boldsymbol{X}_0}^{\boldsymbol{X}_t} \boldsymbol{f}_{inner}\, d\boldsymbol{X} + \int_{\boldsymbol{X}_0}^{\boldsymbol{X}_t} \boldsymbol{f}_{outer}\, d\boldsymbol{X} = \int_{\boldsymbol{X}_0}^{\boldsymbol{X}_t} \boldsymbol{f}\, d\boldsymbol{X}$$
$$= \boldsymbol{f}\boldsymbol{X}_1 + (-\boldsymbol{f}\boldsymbol{X}_2)$$
$$= 0.$$

Therefore we have:

$$\int_{\boldsymbol{X}_0}^{\boldsymbol{X}_t} \boldsymbol{f}_{inner}\, d\boldsymbol{X} = -\int_{\boldsymbol{X}_0}^{\boldsymbol{X}_t} \boldsymbol{f}_{outer}\, d\boldsymbol{X}$$

*i.e.*,

$$\boldsymbol{W}_{inner} = -\boldsymbol{W}_{outer}. \tag{30}$$

Therefore, to calculate the work of outer forces, we can calculate the work of inner forces.

$$\boldsymbol{W}_{outer} = -\boldsymbol{W}_{inner}$$
$$= \int_{\boldsymbol{X}_0}^{\boldsymbol{X}_0 + \Delta \boldsymbol{X}} \boldsymbol{f}_{inner}\, d\boldsymbol{X}$$
$$= \int_{\boldsymbol{X}_0}^{\boldsymbol{X}_0 + \Delta \boldsymbol{X}} k(\boldsymbol{X} - \boldsymbol{X}_0)\, d\boldsymbol{X}$$
$$= \frac{1}{2}k\Delta \boldsymbol{X}^2$$

The energy model is trained to predict the energy of $\frac{1}{k}E(C)$. Therefore, the energy of a complex $C$ is $kE_\theta(C)$. For energy conservation, we have the work of outer forces equals the energy changes as follows:

$$kE_\theta(\tilde{C}) - kE_\theta(C) = \boldsymbol{W}_{outer} = \frac{1}{2}k\Delta \boldsymbol{X}^2,$$

*i.e.*,

$$E_\theta(\tilde{C}) - E_\theta(C) = \frac{1}{2}\Delta \boldsymbol{X}^2,$$

Therefore, the loss for energy conservation is:

$$\mathcal{L}_e = \mathbb{E}[\|E_\theta(\boldsymbol{A}, \tilde{\boldsymbol{X}}) - E_\theta(\boldsymbol{A}, \boldsymbol{X}) - \frac{1}{2}\Delta \boldsymbol{X}_L^2\|^2]$$

### A.4. Force score matching

The force score matching follows the score matching of score-based diffusion model (Dittrich et al., 2018). The object of score matching is

$$\arg\min_\theta \frac{1}{2}\mathbb{E}_{p_{\text{data}}(x)}\left[\|s_\theta(x) - \nabla_x \log p_{\text{data}}(x)\|^2\right]$$

where $s_\theta(x)$ is the prediction of model .i.e. $\tilde{\boldsymbol{f}}_{inner}$ in our model. $\nabla_x \log p_{\text{data}}(x)$ is the derivatives of data distributions. The distribution of $\boldsymbol{f}_{inner}$ in our work is approximately a Gaussian distribution as $\boldsymbol{f}_{inner} \sim N(0, \sigma^2)$. Therefore, we have:

$$p(\boldsymbol{f}_{inner}) = \frac{1}{\sqrt{2\pi\sigma^2}}\exp\left(-\frac{\boldsymbol{f}_{inner}^2}{2\sigma^2}\right)$$

Taking the natural logarithm of $p(\boldsymbol{f}_{inner})$ gives:

$$\log p(\boldsymbol{f}_{inner}) = \log\left(\frac{1}{\sqrt{2\pi\sigma^2}}\right) - \frac{\boldsymbol{f}_{inner}^2}{2\sigma^2}$$

$$= -\frac{1}{2}\log(2\pi\sigma^2) - \frac{\boldsymbol{f}_{inner}^2}{2\sigma^2}.$$

Then take the derivative $\boldsymbol{f}_{inner}$ of $\log p(\boldsymbol{f}_{inner})$ and have:

$$\frac{\partial}{\partial \boldsymbol{f}_{inner}}\log p(\boldsymbol{f}_{inner}) = -\frac{\boldsymbol{f}_{inner}}{\sigma^2}.$$

Therefore, we have the object:

$$\mathcal{L}_{\boldsymbol{f}} = \mathbb{E}[\|\tilde{\boldsymbol{f}}_{inner} - \nabla_{\boldsymbol{f}_{inner}}\log p(\boldsymbol{f}_{inner})\|^2]$$

$$= \mathbb{E}[\|\frac{\partial kE_\theta(\boldsymbol{A}, \tilde{\boldsymbol{X}})}{\partial \tilde{\boldsymbol{X}}_{\text{L}}} + \frac{\boldsymbol{f}_{inner}}{\sigma^2}\|^2]$$

$$= \mathbb{E}[\|\frac{\partial kE_\theta(\boldsymbol{A}, \tilde{\boldsymbol{X}})}{\partial \tilde{\boldsymbol{X}}_{\text{L}}} + \frac{k\Delta\boldsymbol{X}_{\text{L}}}{\sigma^2}\|^2]$$

$$= k\mathbb{E}[\|\frac{\partial E_\theta(\boldsymbol{A}, \tilde{\boldsymbol{X}})}{\partial \tilde{\boldsymbol{X}}_{\text{L}}} + \frac{\Delta\boldsymbol{X}_{\text{L}}}{\sigma^2}\|^2]$$

Therefore, for the total loss of training, the scaler $k$ can be ignored.

## B. Training Details

### B.1. Hardware

We train CEBind, Gauss DSMBind, and DSMBind for 10 epochs. All our experiments are conducted on a computing cluster with 8 GPUs of NVIDIA GeForce RTX 4090 24GB and CPUs of AMD EPYC 7763 64-Core of 3.52GHz. All the inferences are conducted on a single GPU of NVIDIA GeForce RTX 4090 24GB.

### B.2. Hyper-parameter

We train all the models with the same hyperparameters following DSMBind (Jin et al., 2024). The batch size, learning rate, and hidden vector size are 4, 1e-3, and 256, respectively. We use the pre-trained ESM of version esm2_t36_3B_UR50D for protein residue embedding. We use the SRU (Lei et al., 2017) as our protein-ligand interaction modeling model following DSMBind. We assign the duration of $\Delta t$ as a random number from 0 to 1.

## C. Additional Results

### C.1. Results with Different Random Seed

The Pearson correlation on the antibody-antigen and protein-small molecule datasets with different random seeds is shown in Table. 5, where "-" denotes that no positive Pearson correlation is available with this random seed. The results of DSMBind

*Table 5.* Pearson correlation on antibody-antigen and protein-small molecule dataset with different random seeds. Gauss DSM indicates the Gaussian DSMbind method. Mean and dev denote the average value and standard deviation, respectively.

| | antibody-antigen | | | protein-small molecule | | |
| seed | DSMBind | Gauss DSM | CEBind | DSMBind | Gauss DSM | CEBind |
| --- | --- | --- | --- | --- | --- | --- |
| 0 | 0.3551 | 0.2619 | 0.3405 | 0.5937 | 0.5747 | 0.6489 |
| 1 | 0.3273 | 0.3381 | 0.3478 | 0.6455 | 0.6248 | 0.6510 |
| 2 | - | 0.3873 | 0.3724 | - | 0.6328 | 0.6441 |
| 3 | 0.2943 | 0.3249 | 0.3360 | 0.6061 | 0.4877 | - |
| 4 | 0.3572 | 0.3431 | 0.3384 | 0.6200 | 0.4781 | - |
| 5 | 0.3595 | 0.3639 | 0.3365 | 0.3397 | 0.1667 | 0.6431 |
| 6 | 0.3736 | 0.3339 | 0.3423 | 0.6453 | 0.6376 | 0.6467 |
| 7 | 0.3190 | 0.3207 | 0.2347 | 0.6249 | 0.4780 | 0.6457 |
| 8 | 0.2897 | 0.3356 | 0.3556 | 0.6388 | 0.6358 | 0.6456 |
| 9 | 0.3433 | 0.3465 | 0.3249 | 0.6327 | 0.5630 | 0.6461 |
| 10 | - | 0.3402 | 0.3830 | 0.6422 | 0.3282 | 0.6494 |
| 11 | 0.3315 | 0.3389 | 0.3784 | 0.6470 | 0.5366 | 0.6629 |
| 12 | 0.3821 | 0.2402 | - | 0.6066 | 0.6492 | 0.5312 |
| 13 | 0.3515 | 0.3595 | 0.3784 | - | 0.4507 | 0.6417 |
| 14 | - | 0.3171 | 0.3474 | 0.6039 | 0.2564 | 0.6273 |
| Top-5 mean | 0.3655 | 0.3600 | **0.3735** | 0.6438 | 0.6361 | **0.6518** |
| Top-5 dev | 0.0105 | 0.0157 | 0.0096 | 0.0029 | 0.0079 | 0.0052 |
| Top-10 mean | 0.3500 | 0.3487 | **0.3584** | 0.6309 | 0.5820 | **0.6483** |
| Top-10 dev | 0.0190 | 0.0160 | 0.0168 | 0.0149 | 0.0609 | 0.0054 |

and Gauss DSMBind are produced through their official codes. CEBind is much better than previous models in terms of the Pearson correlation on both antibody-antigen datasets and protein-small molecule datasets. Besides, CEBind is more stable than DSMBind, as shown in Table 5: the standard deviation of DSMbind is smaller than DSMBind. Fig. 5 shows the robustness of DSMBind over previous models intuitively.

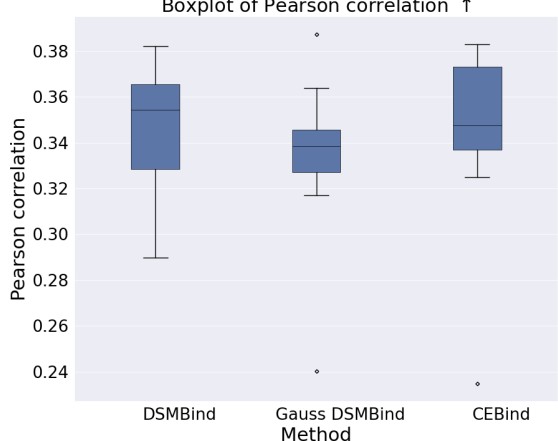

*Figure 5.* Boxplot of Pearson correlation on the antibody-antigen datasets.

*Figure 6.* Boxplot of Pearson correlation on the protein-small molecule dataset.

---

**Algorithm 2** Training procedure of CEBind (single data point)

---

**Require:** A protein-ligand complex $C(\boldsymbol{A}, \boldsymbol{X})$

1: Sample a noise level $\sigma$.
2: Sample forces for each atom in ligand $\boldsymbol{f} \sim \mathcal{N}(0, \sigma^2 \boldsymbol{I})$.
3: Calculate the translation of each atom $\Delta \boldsymbol{X}_{\mathrm{L}}$ with the forces via Eq. 10.
4: Calculate the work of outer forces $\boldsymbol{W}$ via Eq. 12.
5: Apply the noise to the original complex to get the perturbed complex $\hat{C} = (\boldsymbol{A}, \tilde{\boldsymbol{X}})$ via Eq. 11.
6: Predict the energy of both complexes $E_\theta(\boldsymbol{A}, \boldsymbol{X})$ and $E_\theta(\boldsymbol{A}, \tilde{\boldsymbol{X}})$ via energy model.
7: Minimize the energy loss $\mathcal{L}_{\mathrm{e}}$ and force matching loss $\mathcal{L}_{\mathrm{f}}$ from Eq. 13 and Eq. 16.

---

**Algorithm 3** Training procedure of DSMBind (single data point)

---

**Require:** A protein-ligand complex $C(\boldsymbol{A}, \boldsymbol{X})$

1: Sample a noise level $\sigma$, rotation vector, and translation vector for the whole rigid.
2: Apply the noise to original complex to get the perturbed complex.
3: Predict the energy of the perturbed complex.
4: Calculate the gradient of the energy with respect to the atom coordinate.
5: Calculate the rotation of the rigid with the gradient.
6: Minimize DSM objective.

---

*Table 6.* Performance comparison of different training datasets

| Dataset | protein-small molecule | antibody-antigen |
|---|---|---|
| Separately | $0.648 \pm 0.003$ | $0.346 \pm 0.004$ |
| Cross Val | $0.646 \pm 0.001$ | $0.320 \pm 0.002$ |
| Union | $0.646 \pm 0.004$ | $0.352 \pm 0.007$ |

*Table 7.* Pearson correlation on protein-small molecule and antibody-antigen datasets with pre-training on protein-protein pair dataset.

| Pre-train | Freeze | Protein-molecule($r_p \uparrow$) | Antibody-antigen($r_p \uparrow$) |
|---|---|---|---|
| ✓ | ✓ | $0.646 \pm 0.002$ | $0.361 \pm 0.013$ |
| ✓ | | $0.646 \pm 0.002$ | $0.375 \pm 0.014$ |
| | | $0.652 \pm 0.005$ | $0.374 \pm 0.009$ |

## C.2. Comparison between DSMBind and CEBind

The training procedures of CEBind and DSMBind are shown in Algorithm. 2 and Algorithm. 3, where the blue color denotes that the process does not require a gradient, while the red color denotes that the process requires a gradient. The process of calculating the dynamics of the ligand requires a gradient in DSMbind. While CEBind does not require a gradient for the rigid dynamic calculation process, the process can be done in the data loader, which can not only cut down on GPU memory consumption but also accelerate the training process.

## C.3. Generalization

The distribution of the antibody-antigen training set and protein-small molecule test set is shown in Fig. 7. The distribution of the antibody-antigen test set and protein-small molecule training set is shown in Fig. 8. The size range of samples in the antibody-antigen training set encompasses almost all the sample sizes in the protein-small molecule test set. However, the size ranges of the protein-small molecule training set and the antibody-antigen test set overlap only marginally. This is the reason why the CEBind$_{\mathrm{atom}}$ trained on the antibody-antigen training set can generalize to the protein-small molecule test set, while the CEBind$_{\mathrm{atom}}$ trained on the protein-small molecule training set cannot generalize to the antibody-antigen test set. When trained on the union of antibody-antigen dataset and protein-small molecule dataset, the results on antibody-antigen are improved compared with the training on each dataset separately, while the results on protein-small molecule vary a little as shown in Table. 6. The variation in the results is due to the distribution of training data and test data as shown in Fig. 9.

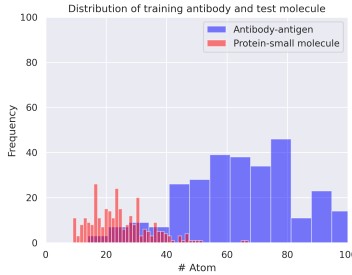

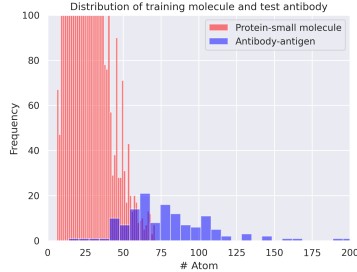

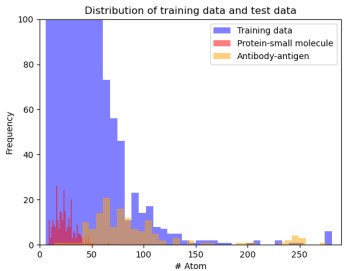

*Figure 7.* The distribution of the number of atoms in the training antibody and test molecules data.

*Figure 8.* The distribution of the number of atoms in the training molecules and test antibody data.

*Figure 9.* The distribution of the number of atoms in the union of the protein-small molecule and antibody-antigen training dataset and test data.

## C.4. Visualization

The heatmap of epitope and paratope binding is shown in Fig. 10 where each pixel indicates the binding energy and the darker the higher. The Spearman correlation between experimental binding energy affinity and predicted binding energy of the Equibind dataset is shown in Fig. 11. The heatmap of pocket and small molecule binding is shown in Fig. 12. The Spearman correlation between experimental binding energy affinity and predicted binding energy on the SAbDab dataset is shown in Fig. 13.

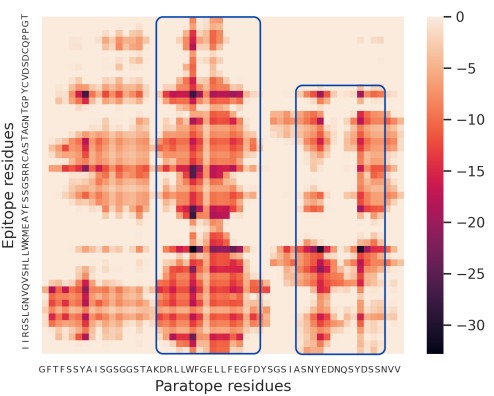

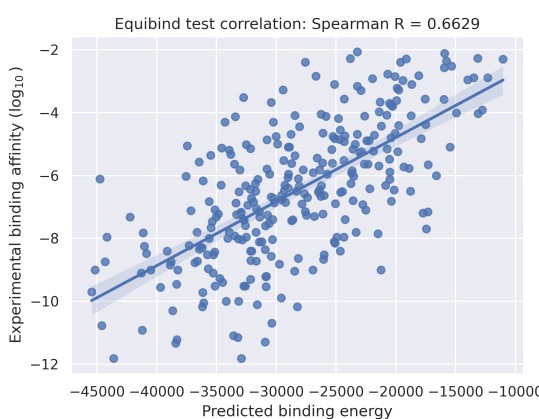

*Figure 10.* The heatmap of the epitope and paratope binding.

*Figure 11.* The Spearman correlation between experimental binding energy affinity and predicted binding energy of the Equibind dataset.

## C.5. Pre-train CEBind with large-scale protein dataset

We try to pre-train CEBind with a large-scale protein dataset. The protein pre-training dataset contains 27,692 protein-protein pairs. The results are shown in Table 7, where freeze denotes that the parameters of the encoder are frozen in training. With pre-training, the Pearson correlation on protein-small molecule gets even worse and the results on antibody-antigen get slightly better. For the protein-small molecule dataset, the pre-trained model with a frozen encoder performs worse than the model without pertaining for two reasons: (1) the model is pre-trained on the protein-protein pairs instead of the protein-small molecule. (2) the parameters of the encoder are frozen, which limits the learning ability of the model. With the pre-trained model as the initial parameters, the model without freezing the encoder parameters still performs worse, which indicates that the model with a bad initialization would also trapped in the local optimal. For the antibody-antigen dataset, the pre-trained model performs better without the encoder parameters freezing. The better performance owes to the

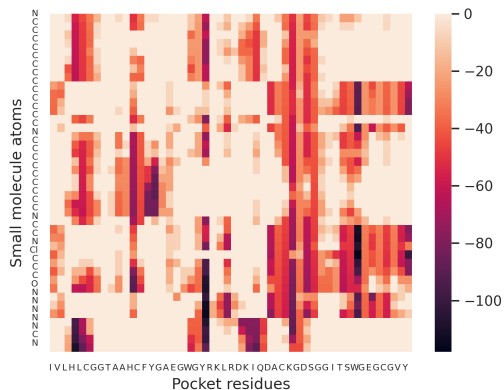

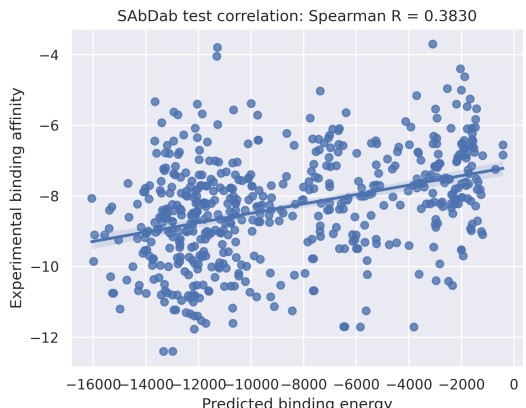

*Figure 12.* The heatmap of the Pocket and Small molecule binding.

*Figure 13.* The Spearman correlation between experimental binding energy affinity and predicted binding energy on the SAbDab dataset.

knowledge obtained through the protein-protein pair pre-training.

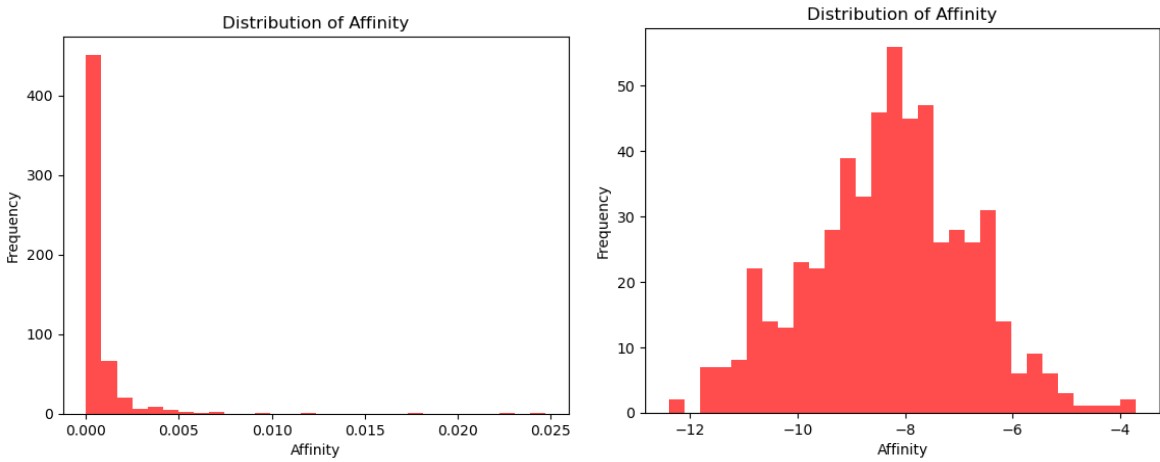

*Figure 14.* The distribution of affinity of original SAbDab dataset *Figure 15.* The distribution of log affinity of SAbDab dataset

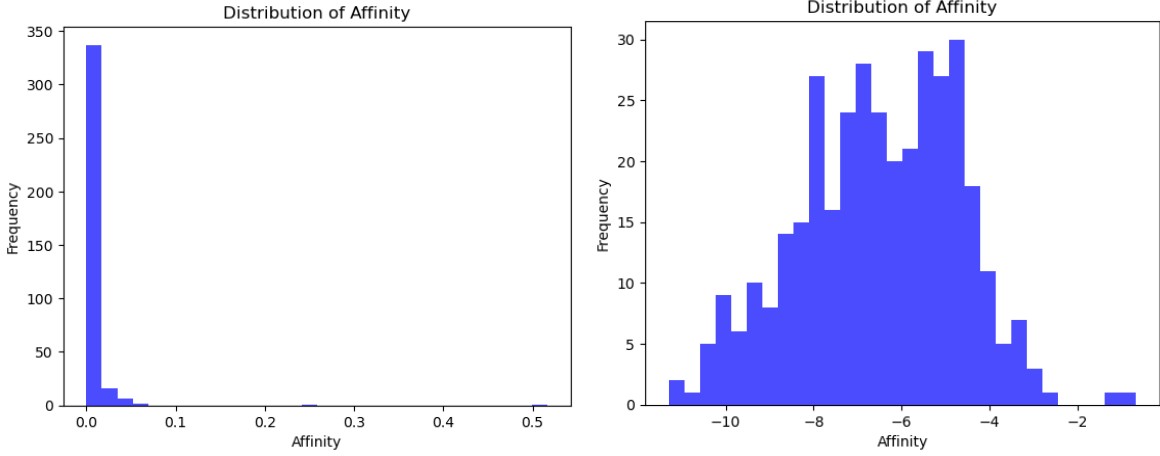

*Figure 16.* The distribution of affinity of original Equibind dataset *Figure 17.* The distribution of log affinity of Equibind dataset

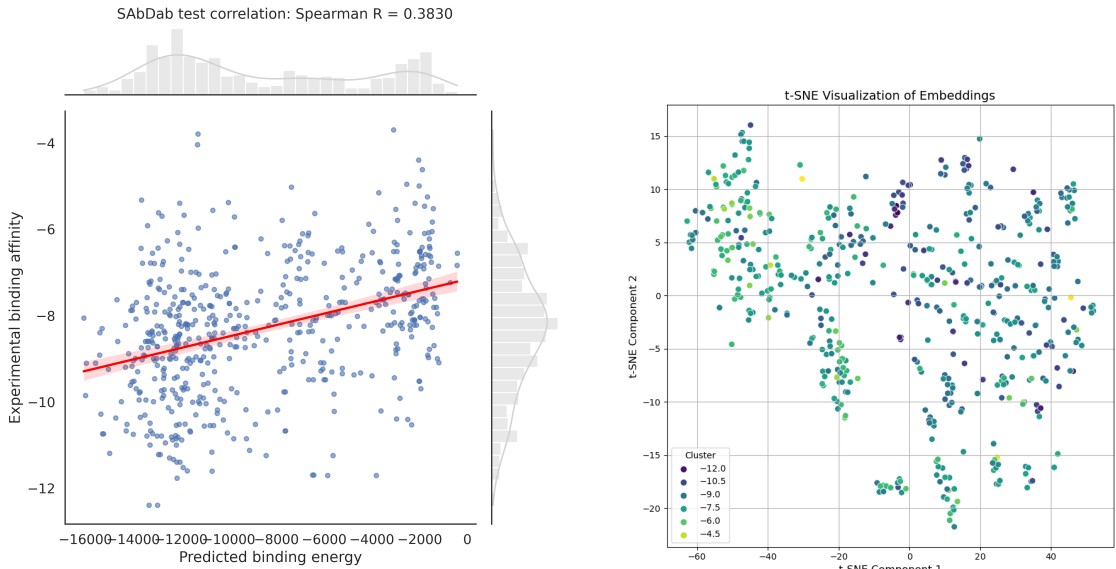

*Figure 18.* The distribution of affinity and energy on SabDab dataset

*Figure 19.* T-SNE visualization of embedding, where colors indicate binding affinity. As shown, there appear to be two clusters. The left one indicates high affinity and the right one low affinity.

