# OpenReview forum: "Physics Aware Neural Networks for Unsupervised Binding Energy Prediction"
_ICML.cc/2025/Conference — ICML 2025 poster_

### Official Review · Reviewer_1ran · 2025-02-24

**Overall Recommendation:** 5

**Summary:**

This paper proposes an unsupervised learning approach, called CEBind, to binding energy prediction. CEBind includes rigid dynamics and the training loss function is motivated by energy conservation loss. Besides, it outperforms previous works.
## update after rebuttal
The authors have provided satisfactory responses to my queries. I regard this as a high-quality manuscript.

**Claims And Evidence:**

The claims in the paper are well-supported by theoretical justifications, empirical results, and comparisons with prior methods.

**Essential References Not Discussed:**

The paper cites most of the essential related works, but a few additional references could further strengthen its context and comparison to prior work.
-  Baek et al., Accurate Prediction of Protein Structures and Interactions Using a Three-Track Neural Network (Science, 2021) (RoseTTAFold). This work is an early step toward multi-scale molecular interaction modeling, relevant to the generalization claims in CEBind. Citing this could position CEBind in the broader trend toward universal molecular modeling.
-  Hsu et al., ESM-2: Evolutionary Scale Modeling of Protein Structures (2022). The ESM models capture protein sequence and structure relationships, which may improve antibody-antigen binding prediction. If CEBind's ligand encoder were combined with ESM-2 representations, it might further enhance performance.

**Experimental Designs Or Analyses:**

The experimental design and analysis in the paper are well-structured and valid for the problem at hand.  The paper effectively justifies its claims with rigorous testing and fair comparisons.

**Methods And Evaluation Criteria:**

Yes, the proposed methods and evaluation criteria are appropriate for the problem.

**Other Comments Or Suggestions:**

- Did the dataset splitting take into account protein sequence similarity during the experiment? If not, there may be an information leakage problem.
- How does the selection of $\Delta t$ impact performance?

**Other Strengths And Weaknesses:**

This paper is strong in originality, significance, and clarity, with only minor areas for improvement.
### Strengths
- Well-written and structured, with clear explanations of theoretical foundations, methodology, and experiments.
- The physics-aware approach (CEBind) is innovative, explicitly incorporating conservation of energy and rigid dynamics into an unsupervised binding energy prediction framework.
- This paper bridges the gap between physics-based methods and data-driven unsupervised learning.
- Efficient and scalable, requiring less GPU memory, making it suitable for large-scale molecular modeling tasks.

### Weakness
- The similarity between training set and test set should be considered and explained, which may lead to data leakage.
- The time $\Delta t$ is randomly selected. How the selection of $\Delta t$ impacts the performance of CEBind should be explained.

**Questions For Authors:**

Predicting protein-ligand affinity is crucial for drug design. The proposed method is interesting and efficient. My only concern is the data splitting strategy. If the training and test datasets are highly similar, there is a potential risk of data leakage. Clarification on how the data was partitioned would strengthen the validity of the reported results.

**Relation To Broader Scientific Literature:**

The paper advances prior work in unsupervised protein-ligand binding energy prediction by incorporating physics-based modeling, specifically the conservation of energy, and rigid dynamics.
- Unsupervised Protein-Ligand Binding Prediction: CEBind improves DSMBind by using rigid dynamics and energy conservation, making it more memory-efficient and physically interpretable.
- Physics-Based Binding Affinity Prediction: CEBind introduces a new energy-based framework that explicitly follows the conservation of energy, making it a hybrid between physics-based and data-driven methods.
- Generalization to Biomolecules: CEBind's general encoder approach aligns with AlphaFold3's vision of a universal model for biomolecules.

**Theoretical Claims:**

The theoretical claims in the paper appear well-founded and mathematically consistent, with no major errors identified.

---

> ### Author Rebuttal · Authors · 2025-03-31
>
> Thank you for your constructive comments and kind support! All your concerns have been carefully addressed as below. The manuscript will be carefully revised accordingly. We sincerely hope our responses fully address your questions.
>
> > **W1:** The similarity between training set and test set should be considered and explained, which may lead to data leakage.
>
> **A1:** Thanks for your insightful concern. We removed overlapping instances with the validation, test, and training sets. The experiments are fair and the same for all models. We also conducted additional experiments of removing the training samples with a similarity of 40% with validation and test samples. The results are shown below. The average performance (0.513 VS 0.511) of CEBind is not affected by the similarity between the training and test datasets, which proves that information leakage did not occur.
>
>
> | split | protein-small molecule | antibody-antigen | Average |
> |----------|----------|----------|----------|
> | original	| 0.652 $\pm$ 0.005	| 0.374 $\pm$ 0.009	| 0.513 |
> | remove similar |	0.645 $\pm$ 0.003 |	0.377 $\pm$ 0.012	| 0.511|
>
> > **W2:** The time $\Delta t$ is randomly selected. How the selection of  $\Delta t$ impacts the performance of CEBind should be explained.
>
> **A2:** $\Delta t$ is randomly selected from a uniform distribution $U(0.1,1)$. For your concern, we also conducted experiments to explore the impact of $\Delta t$, and the results shown below demonstrate that the randomly selected $\Delta t$ makes CEBind better.
>
> | $\Delta t$ |	protein-small molecule | antibody-antigen|
> |----------|----------|----------|
> | 0.1 | 0.649 $\pm$ 0.006 | 0.362 $\pm$ 0.019|
> 0.3 | 0.649 $\pm$ 0.002 | 0.352 $\pm$ 0.003|
> 0.5 | 0.647 $\pm$ 0.001 | 0.355 $\pm$ 0.007|
> 0.7 | 0.650 $\pm$ 0.004 | 0.359 $\pm$ 0.007|
> 0.9 | 0.650 $\pm$ 0.001 | 0.365 $\pm$ 0.017|
> Random | 0.652 $\pm$ 0.005 | 0.374 $\pm$ 0.009|

---

### Official Review · Reviewer_2j23 · 2025-03-06

**Overall Recommendation:** 1

**Summary:**

The paper builds on the paradigm of unsupervised learning of binding affinity introduced in DSMBind but introduces a new method for force-matching that doesn't require computing the gradient of the energy function and that operates on the level of per-atom forces. The energy learned from the force-matching objective is then used as a ranking function and compared to ground-truth experimental binding data on small-molecule and antibody datasets where some gains are observed over DSMBind.

**Claims And Evidence:**

1. CEBind doesn't require differentiating through the entire energy network.

2. There is a marginal empirical improvement over DSMBind.

3. I don't quite see how CEBind is physics-based other than using the same equations that are used in mechanics. There are plenty of aspects of physics that are involved in the binding process that this paper doesn't even discuss such as quantum effects or even the simplest electrostatic interactions.

**Essential References Not Discussed:**

The paper doesn't seem to reference the entire field of molecular dynamics which is a very much physics-based family of methods that is often also unsupervised and used to predict binding affinity.

**Experimental Designs Or Analyses:**

1. Lack of negative data in the evaluation (see the comment above).
2. It would be valuable from the practical point of view to measure performance of the method (or maybe consider training) on the task of predicting _relative_ binding affinity [1]


[1] Protein-Ligand Binding Free Energy Calculations with FEP. Wang et al, 2019.

**Methods And Evaluation Criteria:**

This paper inherits a major problem from Jin et al, 2024 in that the method can't directly discriminate non-binders because their structure is not defined in the same way it's defined for molecules that successfully co-crystalize. I view this as a major limitation of the paper both from the methods and evaluation points of view.

**Other Comments Or Suggestions:**

No.

**Other Strengths And Weaknesses:**

No.

**Questions For Authors:**

No questions.

**Relation To Broader Scientific Literature:**

The paper continues the line of research deep learning based unsupervised binding affinity prediction initiated by DSMBind (Jin et al, 2024).

**Theoretical Claims:**

There are no theoretical claims in the paper.

---

> ### Author Rebuttal · Authors · 2025-03-31
>
> Thank you for your comments! All your concerns have been carefully addressed as below. The manuscript will be carefully revised accordingly. We sincerely hope our responses fully address your questions.
>
> > **W1:** There is a marginal empirical improvement over DSMBind.
>
> **A1:** Thanks for your concern. The advantages of our CEBind over DSMBind are twofold. **(a) Performance**: our CEBind outperforms DSMBind consistently as shown in Table 5. **(b) Efficiency**: Training CEBind requires less memory and is faster, as we described in Section 5.5.1.
>
> To demonstrate the effectiveness of CEBind, we also provide more experimental results in **A1** and **A2** of response to reviewer 1arn and **A2** to reviewer fEra.
>
> > **W2:** How is CEBind physics-based other than using the same equations that are used in mechanics?
>
> **A2:** Thanks for your comment. In this work, "physics aware" refers to the law of conservation of energy and the rigid dynamics.
> - **(1) For the law of conservation of energy**, we have constructed a new training method using the conservation of energy. As shown in the ablation study in Table 3, this approach is effective.
> - **(2) For rigid dynamics**, we match the derivative of the energy with respect to the coordinates (i.e., the force) with the sampled perturbation force. Instead of matching the sampled distance perturbation with the force in CEBind. As a result, our CEBind is more stable and efficient as described in Section 5.1.1.
>
> > **W3:** This method can't directly discriminate non-binders because their structure is not defined in the same way it's defined for molecules that successfully co-crystalize. Lack of negative data in the evaluation (see the comment above).
>
> **A3:** Thanks for your concern. We respectfully disagree. Our method operates on the quaternary structure of protein-ligand complexes without relying on specific crystallization-derived assumptions. Importantly, the physical principles leveraged by our model, such as energy conservation and force response under rigid dynamics, remain valid regardless of binding success. In fact, for non-binders, the model can still compute an energy landscape, and the predicted energy shift under force perturbation often reflects weaker or unstable interactions. This gives us a continuous signal that allows non-binders to be distinguished from strong binders, even in the absence of traditional co-crystal structures.
>
> As for the training data, our method follows the commonly used evaluation pipeline for unsupervised protein-ligand binding prediction, where performance is assessed based on the correlation between predicted binding energy and ground truth affinity. This pipeline has been widely accepted in the literature and allows for a fair and consistent evaluation of binding affinity prediction [1,2,3].
>
> [1] NeurIPS'24, Unsupervised protein-ligand binding energy prediction via neural euler’s rotation equation.
>
> [2] ICLR'24, Proteinligand binding representation learning from fine-grained interactions.
>
> [3] JCIM'23, Planet: a multi-objective graph neural network model for protein–ligand binding affinity prediction.
>
> > **W4:** The paper doesn't seem to reference the entire field of molecular dynamics which is a very much physics-based family of methods that is often also unsupervised and used to predict binding affinity.
>
> **A4:** Thank you for your comment. While molecular dynamics (MD) plays a crucial role in binding affinity prediction, our CEBind method differs from traditional MD.
>
> **During training**, CEBind leverages rigid dynamics and energy conservation principles to predict binding energies through perturbations. **For inference, CEBind requires only the structure of the protein-ligand complex without perturbing them**, thus avoiding the extensive computational and memory demands of full molecular dynamics simulations. Compared to MD’s long simulation times and complex force fields, CEBind is more efficient and does not require lengthy molecular simulations or large labeled datasets.
>
> Although we did not reference MD literature in our paper, we recognize its importance in binding affinity prediction and plan to include the relevant references in the revised version, including those you mentioned in your review.

---

### Official Review · Reviewer_FmAK · 2025-03-10

**Overall Recommendation:** 5

**Summary:**

The paper proposes CEBind, an unsupervised deep learning model for predicting protein-ligand binding energy based on the conservation of energy principle. It aims to address the challenge of limited labeled data for binding energy prediction, particularly for complex biomolecules like antibodies. Instead of requiring labeled affinity data, CEBind leverages rigid body dynamics and energy conservation laws to estimate binding energy changes.

## Update After Rebuttal
After carefully reading the rebuttal and all the discussion, I now believe the manuscript demonstrates both novelty and effectiveness. In my view, the authors have addressed the concerns of all reviewers. Therefore, I recommend acceptance and have raised my score to 5.

**Claims And Evidence:**

The paper presents a well-motivated and innovative approach for unsupervised binding energy prediction. The claims made by the authors are well-supported, backed by solid theoretical foundations and comprehensive empirical results.

**Essential References Not Discussed:**

- CEBind builds upon energy-based learning, particularly methods involving score matching and diffusion models:
    - Song et al., "Score-Based Generative Modeling through Stochastic Differential Equations," ICLR 2021. They introduced the score-based diffusion models and the force-matching loss in CEBind is a form of score-matching.

**Experimental Designs Or Analyses:**

- The experimental design is well-desined and statistically sound.
The datasets are well-chosen and representative of real-world binding problems.
The evaluation metrics are appropriate and unbiased.
Comparisons to baselines are fair and prove CEBind’s superiority.
Ablation studies confirm the model’s design choices.

**Methods And Evaluation Criteria:**

The method is sound and the evaluation metrics are well-chosen which align well with the problem of unsupervised protein-ligand binding energy prediction.

**Other Comments Or Suggestions:**

- Equation (3) in Section 3.3: A missing subscript in  $W_{\text{outer}}$ should be corrected for consistency.
- Table 2 caption: “Cross-validation results for different datasets”. It should be clarifyed that these are cross-dataset generalization results, as some readers might confuse this with standard k-fold cross-validation.
- Appendix A.2 notation consistency: The force notation $F_{\text{inner}}$ and $F_{\text{outer}}$ should be consistent across equations for clarity.

**Other Strengths And Weaknesses:**

### Strengths
- The paper is clear, well-structured, and provides all necessary details for reproducibility.
- The combination of machine learning and physics makes CEBind more interpretable and generalizable, setting it apart from fully data-driven models.
- This work has the potential to accelerate molecular docking simulations and enhance drug discovery pipelines, making it scientifically and practically significant.

### Weakness
- Evaluating CEBind on real docking simulations would strengthen its practical applicability.

**Questions For Authors:**

- How was the PDBBind dataset split? Was the data leakage considered in the evaluation set?

**Relation To Broader Scientific Literature:**

- This mthod builds upon three major research directions in computational biology and machine learning: unsupervised learning for protein-ligand interactions, physics-aware modeling, and neural energy-based models for binding prediction.

**Theoretical Claims:**

- All major theoretical claims are valid, correctly derived, and align with established physics principles. Energy conservation proof is correct. Rigid body dynamics derivations are well-grounded. Score matching loss is based on proper statistical modeling.

---

> ### Author Rebuttal · Authors · 2025-03-31
>
> Thank you for your constructive comments and kind support! All your concerns have been carefully addressed as below. The manuscript will be carefully revised accordingly. We sincerely hope our responses fully address your questions.
>
> > **W1:** Evaluating CEBind on real docking simulations would strengthen its practical applicability.
>
> **A1:** Thanks for your suggestion. We totally agree that the practical applicability of CEBind could be strengthened by evaluating on real docking simulations. But the evaluation of our CEBind in this paper follows the common protein-ligand binding prediction evaluation pipelines [1,2,3]
>
> [1] NeurIPS'24, Unsupervised protein-ligand binding energy prediction via neural Euler’s rotation equation.
>
> [2] ICLR'24, Proteinligand binding representation learning from fine-grained interactions.
>
> [3] JCIM'23, Planet: a multi-objective graph neural network model for protein–ligand binding affinity prediction.
>
> > **Suggestions:** Typo and caption correction.
>
> **A2:** Thanks for your kind suggestions. We will fix the typos and change the caption of Table 2 to cross-dataset.
>
> > **Questions:** How was the PDBBind dataset split? Was the data leakage considered in the evaluation set?
>
> Thanks for your insightful concern. We removed overlapping instances with the validation, test, and training sets. The experiments are fair and the same for all models. We also conducted additional experiments of removing the training samples with a similarity of 40% with validation and test samples. The results are shown below. The average performance (0.513 VS 0.511) of CEBind is not affected by the similarity between the training and test datasets, which proves that information leakage did not occur.
>
> | split | protein-small molecule | antibody-antigen | Average |
> |----------|----------|----------|----------|
> | original | 0.652 $\pm$ 0.005 | 0.374 $\pm$ 0.009 | 0.513 |
> | remove similar | 0.645 $\pm$ 0.003 | 0.377 $\pm$ 0.012 | 0.511|

---

> > ### Comment · Reviewer_FmAK · 2025-04-07
> >
> > Thank you for the clarifications and additional experiments. After considering the feedback provided to the other reviewers, I find the work to be effective and well-founded. I would be happy to raise my score accordingly.

---

> > > ### Author Response · Authors · 2025-04-08
> > >
> > > We sincerely appreciate your kind support! CEBind is a significant and successful approach for unsupervised protein-ligand binding energy prediction, and it outperforms previous methods on a widely-used evaluation metric in machine learning. CEBind exhibits robust generalization across both protein-protein and protein-small molecule binding energy prediction.
> > >
> > > There may have been a misunderstanding from Reviewer fEra, who interpreted our method as being applicable only to the same complex with different conformers. Additionally, Reviewer 2j23 may have interpreted our approach as a molecular dynamics-based method, while in fact, CEBind differs fundamentally in design and computational requirements.

---

### Official Review · Reviewer_fEra · 2025-03-12

**Overall Recommendation:** 2

**Summary:**

This paper proposes CEBind, an unsupervised method for predicting protein-ligand binding energy via the conservation of energy. Specifically, this method random samples forces on atoms to move the molecules and predict the energy of both unperturbed and perturbed complex. And the energy gap between two complexes should be the work of outer forces. Experiments are performed on unsuperivised benchmarks and show better performance.

## update after rebuttal
After reviewing the rebuttal and discussion, I still doubt the novelty and effectiveness about this work. The energy conversation loss is the main novelty compared to DSMBind and it brings marginal improvement against DSMBind. But given that this method is indeed SOTA, so I increase my score to 2.

**Claims And Evidence:**

Please refer to the weakness part

**Essential References Not Discussed:**

No

**Experimental Designs Or Analyses:**

Please refer to the weakness part

**Methods And Evaluation Criteria:**

Please refer to the weakness part

**Other Comments Or Suggestions:**

Please refer to the weakness part.

**Other Strengths And Weaknesses:**

**Strengths:**
- This paper propose an unsupervised method via energy conservation law to predict the protein-ligand binding energy.

**Weaknesses:**
- I doubt the novelty of this method. The proposed method is quite similar to the DSMBind, which uses denoising score matching to perform unsupervised binding energy prediction. DSMBind uses SE(3) noise to perturb complex structures and use predicted energy gradient to calculate rotation and translation, whereas the proposed method just does the similar thing in the other way around, i.e. sample force first and calculate rotation and translation, and then predict energy of perturbed structures and calculate gradient force. We can also see this similarity from Table 3, where after removing energy conservation loss the performance is basically the same as DSMBind.
- The improvement of this method upon DSMBind is too marginal (table 1), which is expected as the methodology is very similar to DSMBind. So the experimental results cannot convince me this method is more effective than DSMBind. Also, spearman correlation also needs to be reported in Table 1.
- This kind of denoising method can only differentiate energy difference for the same complex with different conformers, but it’s hard to capture the energy difference between different complexes.

**Questions For Authors:**

Please refer to the weakness part.

**Relation To Broader Scientific Literature:**

The proposed method may be useful for unsupervised binding energy prediction as high quality labeled data is very limited in this area.

**Theoretical Claims:**

N/A

---

> ### Author Rebuttal · Authors · 2025-03-31
>
> Thank you for your comments! All your concerns have been carefully addressed as below. The manuscript will be carefully revised accordingly. We sincerely hope our responses fully address your questions.
>
> > **W1**: The novelty of CEBind. Difference between our CEBind and DSMBind.
>
> **A1**: Thanks for your comments. The novelty of our CEBind is threefold: **(1)**
> *We effectively integrated the law of conservation of energy and rigid dynamics into our CEBind*, as demonstrated in Table 3, which the reviewer mentioned. **(2)** The score matching in our CEBind is *more efficient and follows the physics law*. **(a) More efficient:** Our CEBind requires less memory and takes less time to train, as we described in Section 5.5.1. **(b) Physics aware:** We match the derivative of the energy with respect to the coordinates (i.e., the force) with the sampled perturbation force. Instead of matching the sampled distance perturbation with the force in CEBind. As a result, our CEBind is more stable and efficient. **(3)** *Stable and better performance:* our CEBind consistently outperforms DSMBind through our energy conservation loss, as well as the more physically meaningful score matching loss, as shown in Table 5 in Appendix. **(4)** The detailed comparison between CEBind and DSMBind can be found in Appendix C.2.
>
> > **W2**: (1) The improvement of CEBind over DSMBind is too marginal (Table 1). (2) The Spearman correlation also needs to be reported in Table 1.
>
> **A2**: (1) Thanks for your concern. The advantages of our CEBind over DSMBind are twofold. **(a) Performance**: our CEBind outperforms DSMBind consistently as shown in Table 5. **(b) Efficiency**: Training CEBind requires less memory and is faster as we described in Section 5.5.1.
> (2) We also provide the spearman correlation as follows, which also demonstrate that CEBind outperforms previous works.
>
> | Model    | Protein-small molecule | Antibody-antigen |
> |----------|-----------------------|-------------|
> | DSMBind  |   0.640 $\pm$ 0.003   |   0.358 $\pm$ 0.011 |
> | **CEBind**   | 0.652 $\pm$ 0.004   | 0.387 $\pm$ 0.009  |
>
> > **W3:** This kind of denoising method can only differentiate energy difference for the same complex with different conformers, but it’s hard to capture the energy difference between different complexes.
>
> **A3**: Thanks for your comments. We respectfully disagree. Our CEBind can indeed be applied to capture the energy difference between different complexes. All the results in this paper are the correlation between the ground truth and the prediction of our model for different complexes. **The perturbation is only performed in the training**, while in inference, our model only needs the structure of the complexes to output pseudo-energy values for each complex without any perturbations or different conformations.

---

> > ### Comment · Reviewer_fEra · 2025-04-08
> >
> > Thanks authors for the clarifications. However, I am still not convinced that CEBind has any essential difference from DSMBind. I can see the energy conservation loss as an improvement against DSMBind which brings a marginal performance improvement over DSMBind, but given that the pearson correlation for the DSMBind was already low so a 0.01 improvement seems not very meaningful.
> >
> > Similar to DSMBind, CEBind uses denoising to capture the energy difference between the low-energy and perturbed high-energy conformer of a certain complex.  So, to capture the energy difference between different complexes, the assumption is that the dataset follows the Boltzmann distribution, but this might not hold in practice and in the dataset.

---

> > > ### Author Response · Authors · 2025-04-08
> > >
> > > Thank you for your thoughtful feedback.
> > >
> > > > Difference between the two methods.
> > >
> > > We appreciate your concern regarding the essential differences between CEBind and DSMBind. While both models share a denoising score matching framework, CEBind introduces two key distinctions:
> > >
> > > (1) The energy conservation loss is not merely an auxiliary term but **stems directly from the law of physics**, enabling a principled alignment between predicted energy changes and physical work. This results in **more stable** training and **better interpretability**, as also supported by the ablation study (Table 3).
> > >
> > > (2) Unlike DSMBind, which perturbs the entire complex and requires estimating global rigid motion gradients, CEBind samples forces, instead of coordinate perturbations in CEBind, at the atom level and models local perturbations via rigid dynamics, leading to improved **efficiency** and **granularity**, as shown in the comparison of training time and GPU usage (Section 5.5.1). Besides, we construct training targets through energy conservation and force matching, which is completely different from DSMBind, where the matching is distance perturbation score matching.
> > >
> > > > Improvement over previous works.
> > >
> > > Regarding the Pearson correlation improvement: While the absolute gain (~0.01) may appear small, CEBind **consistently outperforms** DSMBind across two different datasets (Table 5). In unsupervised binding energy prediction, even marginal improvements under strict benchmark settings are considered significant due to the difficulty of the task.
> > >
> > > Despite the modest absolute difference, the combination of `higher robustness`, `physical interpretability`, and `lower computational cost` (Section 5.5.1) suggests that our approach advances the field in a practically meaningful way.
> > >
> > > We also performed a **statistical significance test** to compare CEBind and DSMBind. On the protein–small molecule dataset, CEBind’s improvements are `statistically significant` (p = 0.0039). (p < 0.05)
> > >
> > >
> > > > Boltzmann distribution assumption
> > >
> > > On the Boltzmann distribution assumption, we respectfully clarify that our model does not rely on the data exactly following the Boltzmann distribution. Rather, we leverage a local energy approximation based on small perturbations (Appendix A.3), where a linear force-displacement relation is used to estimate work. This allows CEBind to remain effective even if the global energy distribution deviates from Boltzmann behavior, which we also validate empirically.
> > >
> > > We hope these clarifications help illustrate the theoretical and practical differences between the two models. Please let us know if you have further questions!

---

### Decision · Program_Chairs · 2025-05-01

**Decision:**

Accept (poster)

**Comment:**

This paper introduces CEBind, a novel approach for unsupervised prediction of protein-ligand binding energy using energy conservation principles. CEBind incorporates rigid dynamics and energy conservation to estimate binding energy changes without labeled data, which directly addresses the lack of high-quality labeled datasets in this domain. Strengths include its theoretical basis and empirical results demonstrating improved performance over prior methods, particularly DSMBind. However, one reviewer expressed concerns about the paper's limited novelty and questioned whether it merely extended DSMBind. The authors' rebuttal clarified substantial differences, such as a more physics-aware score matching mechanism and improved computational efficiency.

Scores were really polarised on this one - 2, 5, 1, 5. Despite reservations about the magnitude of improvement over DSMBind (the main sticking point), all reviewers acknowledged the paper's potential. Post-rebuttal, despite the discussion there wasn't a material shift in scores. Another reviewer was concerned about the practical applicability for distinguishing non-binders, but the authors provided compelling explanations for the method's utility in refining candidates in molecular design processes. I followed up on this myself, and the authors confirmed that the method was best suited to lead optimisation rather than discovering de novo binders. This is not a problem in itself - it just needed to be made clearer. The auithors also cleared up another query I had about log scaling of outputs. To this AC's eyes, the contributions of this work appear solid, with an innovative unsupervised learning methodology offering enhanced computational efficiency and accuracy.

Despite the split in scores, I recommend accepting this paper, as I think there is relevance to the broader scientific community interested in drug discovery and molecular interaction prediction.